# Kinetics of Drug Molecule Interactions with a Newly Developed Nano-Gold-Modified Spike Protein Electrochemical Receptor Sensor

**DOI:** 10.3390/bios12100888

**Published:** 2022-10-17

**Authors:** Dingqiang Lu, Danyang Liu, Xinqian Wang, Yujiao Liu, Yixuan Liu, Ruijuan Ren, Guangchang Pang

**Affiliations:** 1College of Biotechnology & Food Science, Tianjin University of Commerce, Tianjin 300134, China; 2Tianjin Key Laboratory of Food Biotechnology, Tianjin 300134, China; 3Tianjin Institute for Food Safety Inspection Technology, Tianjin 300134, China

**Keywords:** SARS-CoV-2, spike protein, electrochemical receptor biosensor, linkage allosterism

## Abstract

In March 2020, the World Health Organization (WHO) declared COVID-19 a pandemic, and the spike protein has been reported to be an important drug target for anti-COVID-19 treatment. As such, in this study, we successfully developed a novel electrochemical receptor biosensor by immobilizing the SARS-CoV-2 spike protein and using AuNPs-HRP as an electrochemical signal amplification system. Moreover, the time-current method was used to quantify seven antiviral drug compounds, such as arbidol and chloroquine diphosphate. The results show that the spike protein and the drugs are linearly correlated within a certain concentration range and that the detection sensitivity of the sensor is extremely high. In the low concentration range of linear response, the kinetics of receptor–ligand interactions are similar to that of an enzymatic reaction. Among the investigated drug molecules, bromhexine exhibits the smallest Ka value, and thus, is most sensitively detected by the sensor. Hydroxychloroquine exhibits the largest Ka value. Molecular docking simulations of the spike protein with six small-molecule drugs show that residues of this protein, such as Asp, Trp, Asn, and Gln, form hydrogen bonds with the -OH or -NH_2_ groups on the branched chains of small-molecule drugs. The electrochemical receptor biosensor can directly quantify the interaction between the spike protein and drugs such as abidor and hydroxychloroquine and perform kinetic studies with a limit of detection 3.3 × 10^−^^20^ mol/L, which provides a new research method and idea for receptor–ligand interactions and pharmacodynamic evaluation.

## 1. Introduction

In 2019, a novel coronavirus pneumonia (coronavirus disease 19, COVID-19) caused by severe acute respiratory syndrome coronavirus 2 (SARS-CoV-2) emerged, and this disease continues to be a global healthcare problem [1,2]. On 11 March 2020, the World Health Organization (WHO) declared the COVID-19 outbreak a pandemic [3], and as of 22 June 2022, more than 537,5917,664 confirmed cases and 6,319,395 deaths associated with this disease have been reported worldwide [4]. The main clinical symptom of COVID-19 is fever, with most patients presenting moderate or high fever (>38.0 °C), and a small number presenting low fever (37.3–38.0 °C). Other symptoms include fatigue, dry cough, dyspnea, and asthma [5]. SARS-CoV-2 binds to the membrane protein of angiotensin converting enzyme 2 (ACE2) on the surface of lung epithelial cells through the spike protein on its surface [6]. Upon initiation by transmembrane protein serine 2 (TMPRSS2), the virus enters the host cell via subsequent interactions of different subunits and structural domains of the spike protein [7]. Moreover, new virus particles are synthesized via chemical reactions using the cell’s own amino acids, and lipid molecules [8]. These particles are released outside the cell and infect the surrounding normal cells in the same way, thus resulting in many cells being infected by the virus [9]. When the host’s immune system detects the foreign pathogen, it becomes activated, and a large number of immune cells enter the lung tissue, thus releasing cytokines and creating a cytokine storm that attacks the infected cells. This can eventually lead to pneumonia and even to acute respiratory distress syndrome.

Receptors are a class of special proteins present in the cell membrane or within the cell. They amplify and transmit received signals to the cell interior to induce biological effects. The signals are received via a molecular recognition process that implicates the binding of ligand-cell-secreted chemicals that regulate the physiological activities of specific target cells to the receptors. The binding of ligands to receptors is mediated by hydrogen bonding, ionic bonding, and van der Waals interactions. Research on receptor–ligand interactions mainly focuses on the binding between the receptor extracellular domain and the ligand binding [10,11], which is of great importance for drug screening. To date screening methods for anti-SARS-CoV-2 drugs include high-throughput screening [12,13], molecular dynamics simulations [14,15], and biosensors [16,17]. Naveen Vankadari [18] used molecular docking simulation to confirm that the SARS-CoV-2 spike protein is the drug target of arbidol. To prevent the virus shell from contacting, adhering, and fusing to the cell membrane of the host cell, arbidol binds to hemagglutinin (HA), the main glycoprotein on the cell surface.

Gold-soluble nanoparticles with many active centers on the surface can retain their biological activity and good electrochemical properties after adsorption of biological macromolecules, and are widely used in the immobilization of bioactive molecules [19]. After assembling sulfur cordial and AuNPs-HRP on glassy carbon electrode (GCE), the nanomaterials and enzyme reaction system undergo a series of redox reactions with H_2_O_2_ to convert the weak electrical signal into a strong electrical signal [20]. Horseradish peroxidase, the most common member of peroxidases, can catalyze H_2_O_2_ efficiently and is widely used in the field of biosensors [21]. In this study, a new receptor sensor was successfully developed by immobilizing the spike protein and using AuNPs-HRP as the signal amplification system. The interactions of the immobilized spike protein with Shuanghuangliang oral solution, arbidol, chloroquine diphosphate, lopinavir, ribavirin, bromhexine, and hydroxychloroquine sulfate were quantitatively analyzed, and the efficiency and kinetics of receptor–ligand binding were assessed.

## 2. Materials and Methods

### 2.1. Materials and Instruments

Spike protein, arbidol, chloroquine diphosphate, hydroxychloroquine sulfate, ribavirin, bromhexine, horseradish peroxidase (HRP), and bovine serum albumin (BSA) were purchased from Sigma-Aldrich (Shanghai) Trading Co, Ltd. (Shanghai, China), and Shuanghuanglian oral liquid was obtained from Harbin Pharmaceutical Group Sanjing Pharmaceutical Company Limited (Harbin, China). Sodium citrate and chloroauric acid were purchased from Tianjin Yingdaxigui Chemical Reagent Factory (Tianjin, China). NaOH and H_2_O_2_ were obtained from Tianjin Jindong Tianzheng Fine Chemical Reagent Factory (Tianjin, China). Thionin and glutaric dialdehyde were obtained from Shanghai yuanye Bio-Technology Company Limited (Shanghai, China). Acetic acid and chitosan were purchased from Guangzhou Haoying Chemical Technology Company (Guangzhou, China). All reagents used herein were of analytical grade, and ultrapure water was used in all experiments.

The CHI 660E electrochemical workstation and three-electrode system (glassy carbon electrodes ((GCE, Φ = 3 mm): Ag/AgCl reference electrode and platinum wire (Pt) counter electrode) were bought from Shanghai Chenhua Instrument Co., Ltd. (Shanghai, China), whereas the Millipore Milli-Q Pure Water System was provided by Shanghai Yarong Biochemical Equipment Co., Ltd. (Shanghai, China). The Tecnai G2F20 transmission electron microscopy (TEM) was purchased from Philips and was used for the characterization of gold nanoparticles using 200 KV of acceleration voltage. The Dimension 3100 atomic force microscope (AFM) from Veeco (Plainview, NY, USA) and the Quanta FEG 250 scanning electron microscopy (SEM) from FEI (Hillsboro, OR, USA) were used for the characterization of the sensor assembly process.

### 2.2. Experimental Method

#### 2.2.1. Pretreatment and Electrochemical Characterization of Glassy Carbon Electrodes

The glassy carbon electrodes were polished with α-Al_2_O_3_ slurries of different sizes (1.0 μm, 0.3 μm, 0.05 μm) on chamois in sequence, and washed in an ultrasonic water bath for 30 s after each polishing, repeated 2–3 times, and then the glassy carbon electrodes were cleaned with ultrapure water. The glassy carbon electrodes were activated by cyclic voltammetry in 1 mol/L H_2_SO_4_ solution, and the scanning range of voltage during activation was −1.0–1.0 V, and the scanning rate was 100 mV/s. The scanning was repeated until a stable cyclic voltammetry curve appeared.

The activated glassy carbon electrode was placed in 1 × 10^−^^3^ mol/L K_3_Fe(CN)_6_ solution (containing 0.20 mol/L KNO_3_) for cyclic voltammetry curve scanning with a voltage scan range of −0.1–0.6 V and a scan rate of 50 mV/s.

#### 2.2.2. Preparation of the S Protein Receptor Sensor

Five μL of sulfur Thi-Chit compound solution was added to 2.5 mL of 2% chitosan solution (the chitosan solution used 2 g of chitosan dissolved in 100 mL of 1% volume acetic acid solution). This was stirred until completely dissolved to 0.32 mL of 10% glutaraldehyde solution, mixed well by blowing, then 0.2 mL of 0.01 M thionin solution was added, and finally a 2% volume of acetic acid solution was added to make a total volume of 6 mL. This was mixed well and then the solution was ready to use. The solution was dropcast on the working electrode surface. After drying in the ultra-clean bench, the cyclic voltammetry curve was measured. Then, the electrode was immersed in 0.5 mol/L NaOH solution for 5 min, removed and rinsed three times with ultrapure water, and placed in ultrapure water for 0.5 h after cleaning.The above electrode was dried and self-assembled in nano gold–horseradish peroxidase solution (1 mL of gold nanosol [18] was mixed with 1 mL of 2.0 g/L horseradish peroxidase, stirred for 2 h with a magnetic stirrer and allowed to stand for 12 h at 4°C before use) at 4 °C for 24 h; its cyclic voltammetry curves were measured after assembly, and the characterization of gold nanoparticles is provided in the attached document.After washing the electrode surface with ultrapure water, the S protein solution was added dropwise on the electrode surface and self-assembled at 4 °C for 24 h. The cyclic voltammetry curve was measured after assembling.The electrode was removed and washed with ultrapure water, finally coated with BSA solution (0.5 g/100 mL) and incubated for 1 h at 37 °C to close the non-specific sites, and their cyclic voltammetry curve was measured. The nano-gold-modified receptor spike protein electrochemical biosensor was obtained, and the assembly process is shown in Figure 1: Characterization of electrode surface morphology during electrode assembly using SEM.

The above steps were performed in 1 × 10^−^^3^ mol/L K_3_Fe(CN)_6_ (containing 0.20 mol/L KNO_3_) solution by cyclic voltammetry with a scanning potential range of −0.1 to 0.6 V and a scanning rate of 50 mV/s, using a three-electrode system; the prepared S protein receptor sensor was used as the working electrode, the Ag/AgCl electrode as the reference electrode, and the platinum wire electrode as the counter electrode. After assembling thi and HRP on GCE, redox reaction occurs with H_2_O_2_: (1): H_2_O_2_ + HRP → Compound Ⅰ + H_2_O; (2): Compound Ⅰ + thi (red) → Compound Ⅱ + thi (ox)*; (3): Compound Ⅱ + thi (ox)* + 2H^+^ → HRP + thi (ox) + H_2_O; (4): thi (ox) + 2e^−^ +2H^+^ → thi (red).

#### 2.2.3. Assessment of the Interaction of the S Protein Receptor Sensor with Related Drugs

Using a three-electrode system, the prepared spike protein receptor sensor was used as the working electrode, the Ag/AgCl electrode as the reference electrode, and the platinum wire as the counter electrodes, ultrapure water was used as the blank control, and the response currents of different concentrations of arbidol, hydroxychloroquine, and other related drugs were measured by time amperometry at a certain voltage. The rate of change of response currents was used as the detection index (H_2_O_2_ was added in advance to reach a final concentration of 8 mmol/L), and calculated according to the following equation:(1)$$∆I=\frac{I_{1}-I_{2}}{I_{1}}$$

Here, *I*_1_ and *I*_2_ represent the pre- and post-measurement values of steady-state current at the same time points as the response current of the related drugs, respectively.

#### 2.2.4. Simulation of Molecular Docking of the Spike Protein with Six Ligands

The crystal structure of the SARS-CoV-2 spike protein was obtained from the PDB protein and the UniProt protein databases, while the structures of the six ligands were acquired from PubChem.

The receptor protein and ligands were processed before molecular docking simulations. First, water molecules and other residue ligands of the S protein were removed using PyMOL. Then, hydrogen atoms (molecular docking acceptors) were added to the six ligands to obtain the molecular docking ligand. Finally, torsion bonds on the ligands were detected.

Simulations of the molecular docking of the spike protein and six ligands were performed using the AutoDock Vina software (Ver. 1.2.3.). This software continuously adjusts the conformation of the ligand molecule (including its orientation, position, and energy) within the Gridbox range and scores the ligand’s different conformations. During the docking process, each of the six ligands produced nine docking conformations that were then ranked from high to low, based on affinity values. The conformation with the highest affinity value achieves the best geometric and energy matching with the receptor protein. Subsequently, the binding modes and interactions of the spike protein with the six ligands were analyzed using PyMol and ligPlus.

## 3. Results

### 3.1. Electrochemical Characterization of Spike Protein Receptor Sensors

Cyclic voltammetry was used to characterize the electrode assembly at each stage in the preparation process, and the modification of biomolecules on the electrode surface was roughly judged based on the size of the redox peaks in the cyclic voltammogram.

As shown in Figure 2a, cyclic voltammetry was performed in 1 × 10^−^^3^ mol/L K_3_Fe(CN)_6_ (containing 0.20 mol/L KNO_3_) solution. After the pretreated electrode surface (curve a) was dropwise added with thionine-chitosan (curve b), the peak current increased rapidly, which was due to the good electron transfer ability of thionine, indicating that the thionine–chitosan composite film was successfully assembled to the electrode surface. Figure 2b shows our previous study [19]; curve one is the cyclic voltammogram of Thi-Chit/GCE and curve two is the cyclic voltammetry of Chit/GCE, both with 1 × 10^−^^3^ mol/L K_3_Fe(CN)_6_ solution containing 0.20 mol/L KNO_3_ as the test substrate (scan range 0.6 to −0.1 V, scan rate 50 mV/s). It can be seen from the figure that the peak current of the Thi/Chit/GCE electrode is significantly larger than that of the Chit/GCE electrode, and the peak current increases gradually when thi is present, indicating that thi facilitates the transfer of electrons between the electrode and the substrate. The curve corresponding to the nano-gold–horseradish peroxidase composite membrane (curve c) exhibits a higher redox peak current than the curve corresponding to thionine–chitosan due to the high electron density and dielectric properties of nano gold. These properties improve the electron transfer efficiency of the material, and this effect exceeds the hindering effect of horseradish peroxidase on electron transport. Therefore, the increased current indicates that the nano-gold–horseradish peroxidase composite membrane was successfully assembled. Compared to curve c, the redox peak current of the curve obtained after self-assembly of the spike protein receptor (curve d) is slightly reduced, probably due to the covalent binding of gold nanoparticles with the thiol-containing amino acids (methionine, cysteine) in the receptor spike protein through Au-S bonds. The presence of protein macromolecules hinders the electron transfer on the electrode surface. Finally, the redox peak current decreases upon dripping with BSA (curve e) due to the effect of this reagent in blocking the non-specific sites. Overall, the results indicate that the sensor was successfully prepared. Before use, the prepared sensor was stored in phosphate buffer at 4 °C.

The scanning electron microscope (SEM) of the sensor assembly process is shown in Figure 3. From Figure 3a, it can be seen that the surface of GCE is smooth and free of other impurities with high finish; Figure 3b is the SEM image of Thi-Chit assembled onto GCE, from which it can be seen that the surface structure of the chitosan membrane is rough and there are incompletely dissolved particles; Figure 3c shows the AuNPs-HRP/Thi-Chit/GCE and it can be seen that AuNPs-HRP is bound to the Thi-Chit membrane in a pine tree shape, which is similar to the TEM results of AuNPs-HRP; when the spike protein is assembled to the AuNPs-HRP membrane (Figure 3d), SEM shows that the spike protein has a tree-like structure, in which can be seen spherical Au particles, which is similar to the TEM results of the AuNPs-HRP/spike protein; Figure 3e shows the SEM image of BSA assembled onto the electrode, where BSA is attached to the electrode surface in a snowflake shape and tightly packed granular spherical proteins appear in different degrees.

### 3.2. Regulation of the Spike Protein Receptor Sensor and Related Drugs

In the detection of the concentration gradient of the drug to be tested, the minimum detection limit was reached when the concentrations of Shuanghuanglian oral liquid, arbidol, and lopinavir were 1 × 10^−^^20^ mol/L, ribavirin and chloroquine diphosphate were 1 × 10^−^^19^ mol/L, hydroxychloroquine was 1 × 10^−^^14^ mol/L, and bromhexine was 1 × 10^−^^21^ mol/L. Since the concentration range of the drug to be tested is wide and not easy to graph, the concentrations were redefined as follows: pC is the redefined concentration of the drug to be tested, both starting from 1, and C is the initial concentration of the drug to be tested. For example, C_Shuanghuanglian oral liquid_ is 1 × 10^−^^20^ mol/L and pC_Shuanghuanglian oral liquid_ is 1, at which point pC_Shuanghuanglian oral liquid_ = 21 + lg(C). Other drug concentrations to be measured are defined by the following equations:pC_Arbidol Hydrochlorrid__e_ = 21 + lg(C)
pC_Chloroquine diphosphate_ = 20 + lg(C)
pC_Lopimavir_ = 21 + lg(C)
pC_Ribavirin_ = 20 + lg(C)
pC_Bromhexine HCl_ = 22 + lg(C)
pC_Hydroxychloroquine Sulfate_ = 15 + lg(C)
pC_KCl_ = 16 + lg(C)

Figure 4 shows plots of the response current change rate versus the logarithmic value of the relevant drug concentration. Meanwhile, different concentrations of KCl solutions were used as blank controls.

### 3.3. Kinetics of the Interaction of the Spike Protein Receptor with Different Drug Molecules

As shown in Figure 5, the rate of change of the response current increases linearly in the Shuanghuanglian oral liquid, arbidol, chloroquine diphosphate, lopinavir, ribavirin, bromhexine, and hydroxychloroquine concentration ranges of 10^−^^20^–10^−^^18^, 10^−^^20^–10^−^^18^, 10 ^−^^19^–10^−^^17^, 10^−^^19^–10^−^^17^, 10^−^^19^–10^−^^17^, 10^−^^21^–10^−^^19^, and 10^−^^14^–10^−^^12^ mol/L, respectively. The detected concentrations were further divided within these concentration ranges, and the current change rates were plotted again as a function of drug concentrations.

The interaction between the receptor and the ligands may be described by the following equation:(2)R+LRL←K2→K1 
with
(3)$$K_{d}=\frac{k_{2}}{k_{1}}=\frac{\left[R\right]\left[L\right]}{\left[RL\right]}\;$$

If [*RT*] designates the initial concentration of the receptor, then [*R*] = [*RT*] − [*RL*]; if [*LT*] is the total ligand concentration, then [*L*] = [*LT*] − [*RL*]. By substituting [*L*] with [*LT*] − [*RL*] and [*R*] with [*RT*] − [*RL*] in Equation (3), then the following hyperbolic quadratic equation is obtained:(4)$${\left[RL\right]}^{2}-\left[RL\right]\left\{\left[RT\right]+\left[LT\right]+K_{d}\right\}+\left[RT\right]\left[LT\right]=0\;$$

Equation (4) has one unknown, [*RL*], and when [*RT*]/*K_d_* is fixed, [*RL*] varies as a function of [*LT*]: it rapidly rises at the beginning, and then gradually stabilizes at a constant value. The [*RL*] versus [*LT*] variation curve corresponds to the saturation curve of the receptor–ligand interaction. Based on Equation (4), the binding of receptors to ligands is governed by a ligand saturation effect, similar to the “substrate saturation effect” observed for enzyme–substrate interactions.

The curves in Figure 5 were fitted by hyperbolic functions using the Origin 2018 software, and the fitted curves are presented in Figure 6.

As shown in Figure 6, in the low concentration range, the rate of current change increases with increasing drug concentration, which indicates that the receptor is not yet saturated. Beyond a certain concentration, the rate of current change remains constant or increases slightly, thus indicating that the receptor has reached saturation.

By rearranging Equation (3), the following double reciprocal equation is obtained:(5)$$\frac{1}{\left[RL\right]}=\frac{1}{\left[RT\right]}+\frac{K_{d}}{\left[RT\right]}\frac{1}{\left[L\right]}$$

The plot of 1/[*RL*] as a function of 1/[*L*] yields a straight line with *K_d_*/[*RT*] slope, −1/*K_d_* horizontal axis intercept, and 1/[*RT*] vertical axis intercept. Figure 7 depicts the straight-line plots obtained for different drug molecules.

The linear regression equations of the different drug molecules tested herein are given below:$$\mathrm{Shuanghuanglian}\;\mathrm{oral}\;\mathrm{liquid}\;(A):\;\frac{1}{∆I}=0.919\times 10^{-20}\frac{1}{C}+1.511\;(R^{2}=0.988)$$
$$\mathrm{Abridol}\;(B):\;\frac{1}{∆I}=1.957\times 10^{-20}\frac{1}{C}+2.408\;(R^{2}=0.990)$$
$$\mathrm{Chloroquine}\;\mathrm{diphosphate}\;(C):\;\frac{1}{\Delta I}=4.597\times 10^{-20}\frac{1}{C}+1.740\;(R^{2}=0.994)$$
$$\mathrm{Lopinavir}\;(D):\;\frac{1}{\Delta I}=0.966\times 10^{-20}\frac{1}{C}+1.528\;(R^{2}=0.987)$$
$$\mathrm{Ribavirin}\;(E):\;\frac{1}{\Delta I}=2.487\times 10^{-19}\frac{1}{C}+1.961\;(R^{2}=0.938)$$
$$\mathrm{Bromhexine}\;(F):\;\frac{1}{\Delta I}=0.723\times 10^{-21}\frac{1}{C}+1.499\;(R^{2}=0.990)$$
$$\mathrm{Hydroxychloroquine}\;(G):\;\frac{1}{\Delta I}=0.839\times 10^{-14}\frac{1}{C}+2.192\;(R^{2}=0.975)$$

Based on the above equations, the allosteric constant Ka values corresponding to interactions with Shuanghuanglian oral liquid, arbidol, chloroquine diphosphate, lopinavir, ribavirin, bromhexine, and hydroxychloroquine are 6.083 × 10^−^^21^, 8.140 × 10^−^^21^, 2.644 × 10^−^^19^, 6.325 × 10^−^^20^, 1.270 × 10^−^^19^, 4.823 × 10^−^^22^, and 3.825 × 10^−^^15^ mol/L, respectively. Notably, the allosteric constant Ka determined herein is similar to the enzymatic reaction parameter Km, which is defined as the ligand concentration at half maximum receptor biological effect. The smaller the Ka value, the higher the biological efficiency of the receptor–ligand interaction.

### 3.4. Molecular Docking Simulation Results

Affinity is an evaluation criterion for the degree of binding of the target receptor to the small-molecule ligand. The higher the absolute value of affinity, the stronger the ligand-receptor binding ability and the more stable the complex formed. The affinity values of the optimal binding conformation determined using molecular docking simulations of the spike protein with lopinavir are listed in Table 1 (see the Appendix A for the remaining values). Six small-molecule drugs exhibit negative affinity values, thus indicating that the reactions of the spike protein with these molecules are spontaneous.

When the docking affinity is −8.9 kcal/mol, a hydrogen bonding interaction occurs between the spike protein and lopinavir, and the RMSD of lopinavir is 2.124. The ASN394 residue interacts with lopinavir via hydrogen bonding (HB) at distances of 2.9 and 3.0 Å, and there are no π-π or π-cation interactions (Figure 8A). Based on LigPlus analysis, residues Trp69, Ala99, Leu73, Leu391, Arg393, Phe390, Asp350, Phe40, Trp349, Ala348, His378, His401, and Glu402 exhibit hydrophobic interactions (HI) with lopinavir.

When the docking affinity is −4.7 kcal/mol, the spike protein and chloroquine diphosphate are bound by hydrogen bonds, and the RMSD of the ligand is 2.075. The hydrogen bonding interaction of the LEU391 residue of the spike protein with chloroquine diphosphate occurs at a distance of 1.9 Å, and there are no π-π or π-cation interactions between the two entities (Figure 8B). The results of LigPlus analysis demonstrate that residues Phe390, Leu73, Lys74, and Gln102 exhibit hydrophobic interactions with chloroquine diphosphate.

When the docking affinity between the spike protein and arbidol is −5.9 kcal/mol, the two entities interact via hydrogen bonding, and the RMSD of arbidol is 3.718. The lengths of the hydrogen bonds binding arbidol to the ASP206, TRP203, and LYS562 residues of the spike protein are 3.1, 2.1, and 3.1 Å, respectively, and no π-π or π-cation interactions exist between the receptor and the ligand (Figure 8C). Residues Tyr202, Asn397, Asn394, Gly205, Glu398, and Tyr196 have hydrophobic interactions with arbidol, as shown by LigPlus analysis.

Hydrogen bonding interaction between the spike protein and hydroxychloroquine sulfate occurs when the docking affinity is −6.5 kcal/mol. The RMSD of the ligand in this complex is 3.172. The hydrogen bonds exist between hydroxychloroquine sulfate and the TYR196, GLU208, GLN102, GLN98, and ASN210 residues of the spike protein. Their lengths are 2.4, 2.2, 3.0, 3.0, and 2.3 Å, respectively. Again, no π-π or π-cation interactions exist between the receptor and the ligand (Figure 8D). Based on LigPlus analysis, residues Val209, Lys562, Gly205, and Leu95 exhibit hydrophobic interactions with hydroxychloroquine sulfate.

When the docking affinity is −7.0 kcal/mol, the spike protein and ribavirin interact via hydrogen bonding. The lengths of the hydrogen bonds binding ribavirin to the ALA396, GLU208, TRP566, GLN98, and ASN210 residues of the spike protein are 2.0, 2.2, 3.1, 2.1, 2.2, and 3.1 Å, respectively, and there are no π-π or π-cation interactions between the receptor and the ligand (Figure 8E). The results of LigPlus analysis demonstrate that hydrophobic interactions occur between ribavirin and the Pro565, Val209, Lys562, and Leu95 residues of the spike protein.

The spike protein and bromhexine are bound by hydrogen bonds when the docking affinity between them is −5.8 kcal/mol. Under this condition, the RMSD of bromhexine is 2.023. The two hydrogen bonds binding the TYR385 and ASP382 residues of the spike protein to bromhexine are 2.2-Å-long, and there are no π-π or π-cation interactions between the two entities (Figure 8F). Residues His401, His378, Ala348, Asp350, and Thr347 exhibit hydrophobic interactions with bromhexine, as shown by LigPlus analysis.

### 3.5. Molecular Docking Simulation and Ka Value

The sensing efficiency and kinetics of the spike protein sensor prepared herein were tested against seven different drugs, such as arbidol and bromohexine, at varying drug concentrations. Specifically, the allosteric constants of the drugs were determined. The results demonstrate that bromhexine has the smallest Ka value, which means that it yields the greatest change in electrochemical signal upon interacting with the receptor; thus, it is most sensitively detected. The detection sensitivity of the drugs decreases in the following order: bromhexine > Shuanghuanglian oral liquid > arbidol > lopinavir > ribavirin > chloroquine diphosphate > and hydroxychloroquine.

By analyzing the interactions between small-molecule ligands and biomacromolecular receptors, and by predicting the binding modes and affinities between the two entities, molecular docking simulations provide important information on structure-based drug design. Affinity is an evaluation criterion for the degree of binding between the target receptor and the small-molecule ligand, and the higher the affinity (absolute value), the stronger the ligand-receptor binding ability and the more stable the complex formed. As discussed in Section 3.4, the molecular docking results reveal that lopinavir exhibits the highest affinity with the prepared receptor sensor (−8.9 kal/mol), followed by ribavirin (−7.0 kal/mol), hydroxychloroquine (−6.5 kal/mol), arbidol (−5.9 kal/mol), bromhexine (−5.8 kal/mol), and chloroquine diphosphate (−4.7 kal/mol). Since the affinity values are all negative, the reactions of the drug molecules with the sensor are spontaneous (Table 2).

## 4. Discussion

Amid the COVID-19 pandemic, doctors and researchers around the world are working hard to find effective treatments and develop antiviral drugs. By preventing the SARS-CoV-2 virus from entering cells, antiviral drugs reduce the speed at which the virus spreads and replicates in the host by preventing the SARS-CoV-2 virus from entering cells. SARS-CoV-2 encodes four structural proteins including a spike glycoprotein, a membrane glycoprotein, an envelope protein, and a nucleocapsid protein [22,23], as well as non-structural proteins such as 3-chymotrypsin-like protease (3CLpro), RNA-dependent RNA polymerase (RdRp), and papain-like protease (PLpro) [24]. Nonstructural proteins, spike glycoproteins, TMPRSS2, and ACE2 are important drug targets for anti-COVID-19 treatment. In this study, spike proteins were assembled on gold nanoparticles to study their interaction with seven antiviral drugs. The kinetics of receptor–ligand binding were assessed, and the Ka values corresponding to interactions with different drug molecules were determined. The results show that bromhexine exhibits the lowest Ka value among the investigated drug molecules and thus, is most sensitively detected by the prepared sensor. Kenhinde et al. [25] investigated the inhibition mechanism of bromohexine and ambroxol hydrochlorides as blockers of the interaction of SARS-CoV-2 spike protein with ACE2 molecules and showed that bromohexine exhibited better molecular interaction and binding than ambroxol hydrochlorides with reduced affinity. This is consistent with the results of the present study in which bromoxyn had the smallest Ka value and the most sensitive sensing. In addition, Table 3 compares the research work on electrochemical sensors based on SARS-CoV-2 diagnosis of COVID-19 in the last three years. Molecular docking simulations revealed that the -NH_2_ group of the bromohexine benzene ring forms hydrogen bonds with the -COOH and -OH groups of the Asp382 and Tyr385 residues of the spike protein, respectively. Both hydrogen bonding interactions occur at a distance of 2.2 Å. Moreover, five pairs of hydrophobic interactions exist between bromhexine and the spike protein. Bromhexine is widely used in the treatment of a series of respiratory diseases and is a selective inhibitor of transmembrane serine protease 2 [26]. In the first step of viral infection, the virion binds to the surface of the host receptor and fuses with the cell membrane. However, only the spike protein, which is activated by proteolysis, can release the fusion peptide into the target cell in a controlled manner [27], resulting in cell infection. The main hydrolase implicated in the viral infection process is TMPRSS2 [28].

## 5. Conclusions

Various approaches have been reported using sensors to detect SARS-CoV-2 virus and to screen potential drugs for the treatment of COVID-19 by therapeutic target analysis, but there are no studies on the kinetics of interactions between the spike protein and antiviral drugs. The spike protein receptor sensor proposed in this study was prepared by immobilizing the spike protein as the measuring element, and then immobilizing thionine and chitosan molecules with strong conductivity and redox properties. The biocompatibility of chitosan and its abundant -NH^3+^ groups promotes the adsorption and fixing of gold nanoparticles that were used to adsorb the spike protein due to their large specific surface area and good electrical conductivity. Horseradish peroxidase was also incorporated into the sensor as a signal amplification system. The efficiency and kinetics of the spike protein interaction with Shuanghuanglian oral liquid, arbidol, chloroquine diphosphate, lopinavir, ribavirin, bromhexine, and hydroxychloroquine were analyzed using the time-current method (I-T method). The electrochemical receptor sensor prepared herein allows for the visualization and kinetic assessment of the interactions between the spike protein and a variety of drugs with a limit of detection 3.3 × 10^−^^20^ mol/L, thus providing a new and simple method for the detection of receptor–ligand interactions at a relatively low cost. In particular, this receptor sensor provides new ideas for studying receptor–ligand interactions. However, the current receptor sensors still have some shortcomings, for example, in the practical application of their characteristics, and final analytical performance in the inter-sensor instability may lead to experimental failure, so the development of electrochemical biosensors suitable for general promotion and use that have high accuracy and stability still needs continuous efforts to promote the receptor sensors gradually towards the direction of functional diversification, miniaturization and integration, where it will show a wider range of applications in the field of disease diagnosis, genetic testing, etc.

## Figures and Tables

**Figure 1 biosensors-12-00888-f001:**
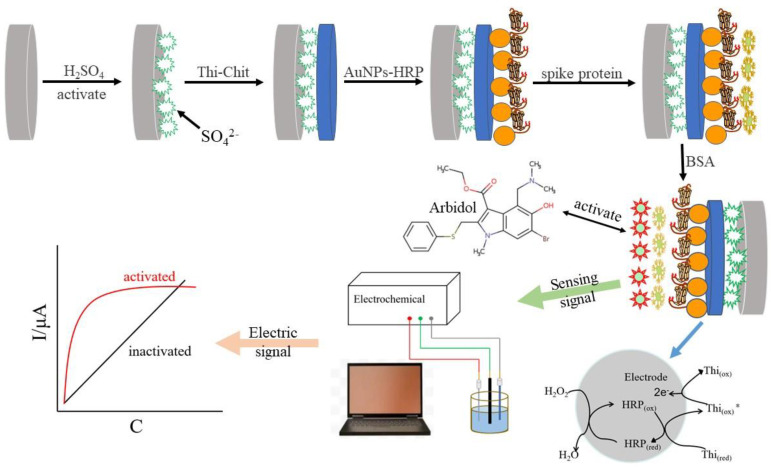
Schematic illustration of the fabrication process of the spike protein receptor biosensor.

**Figure 2 biosensors-12-00888-f002:**
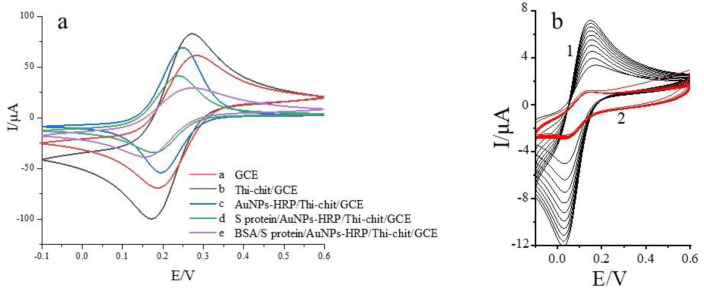
(**a**) Cyclic voltammetry characterization of spike protein receptor sensor electrode assembly modifications. (**b**) the impact of thionine on electrochemical properties characterized by cyclic voltammetry; curve 1 is the cyclic voltammogram of Thi-Chit/GCE and curve 2 is the cyclic voltammogram of Chit/GCE. The peak current increases gradually when thi is present, indicating that thi facilitates the transfer of electrons between the electrode and the substrate. Cyclic voltammetry was performed in 1 × 10^−^^3^ mol/L K_3_Fe(CN)_6_ (containing 0.20 mol/L KNO_3_) solution with a scanning potential range of −0.1 to 0.6 V and a scanning rate of 50 mV/s, using a three-electrode system; the prepared S protein receptor sensor was used as the working electrode, Ag/AgCl electrode as the reference electrode and platinum wire electrode as the counter electrode.

**Figure 3 biosensors-12-00888-f003:**
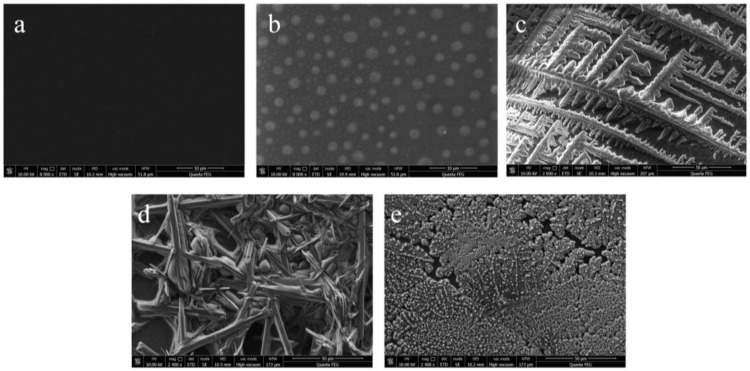
SEM of sensor assembly process: (**a**) GCE; (**b**) Thi-Chit/GCE; (**c**) AuNPs-HRP/Thi-Chit/GCE; (**d**) spike protein/AuNPs-HRP/Thi-Chit/GCE; (**e**) BSA/spike protein/AuNPs-HRP/Thi-Chit/GCE.

**Figure 4 biosensors-12-00888-f004:**
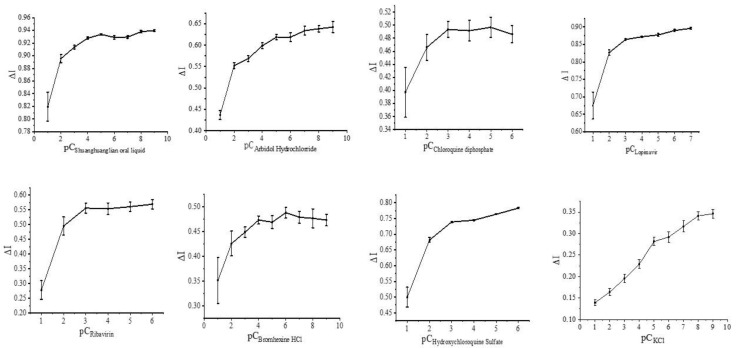
Rate of change of response current in the detection range (the concentration range of Shuanghuanglian oral liquid, arbidol, and lopinavir were 1 × 10^−^^20^ mol/L, ribavirin and chloroquine diphosphate were 1 × 10^−^^19^ mol/L, hydroxychloroquine was 1 × 10^−^^14^ mol/L, bromhexine was 1 × 10^−^^21^ mol/L, and KCl was 1 × 10^−^^15^ mol/L as control).

**Figure 5 biosensors-12-00888-f005:**
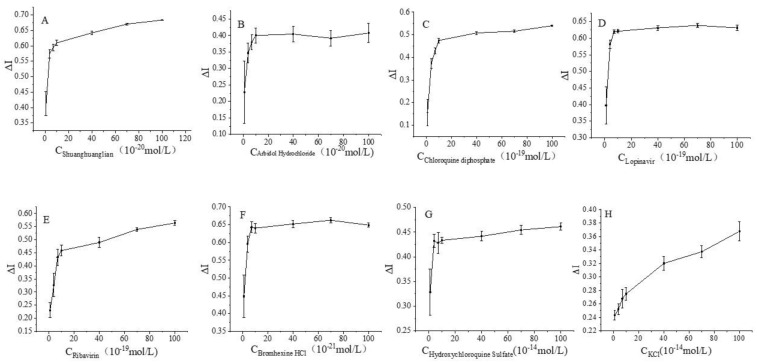
Variation of the rate of change of the response current as a function of drug concentration: (**A**) Shuanghuanglian oral liquid 10^−^^20^–10^−^^18^ mol/L; (**B**) abridol 10^−^^20^–10^−^^18^ mol/L; (**C**) chloroquine diphosphate 10^−^^19^–10^−^^17^ mol/L; (**D**) lopinavir 10^−^^19^–10^−^^17^ mol/L; (**E**) ribavirin 10^−^^19^–10^−^^17^ mol/L; (**F**) bromhexine 10^−^^21^–10^−^^19^ mol/L; and (**G**) hydroxychloroquine 10^−^^14^–10^−^^12^ mol/L; (**H**) KCl 10^−^^15^–10^−^^13^ mol/L as control.

**Figure 6 biosensors-12-00888-f006:**
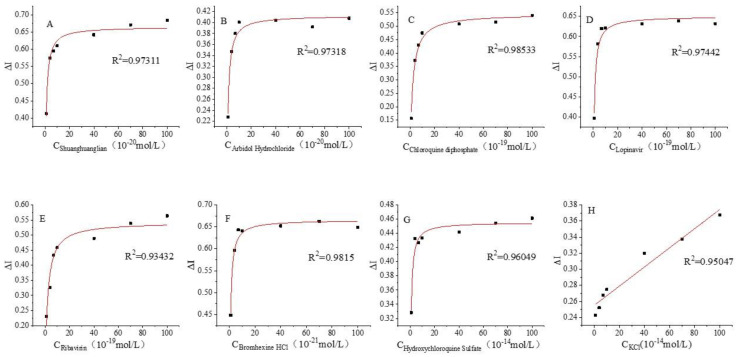
Fitted response curves: (**A**) Shuanghuanglian oral liquid 10^−^^20^–10^−^^18^ mol/L; (**B**) abridol 10^−^^20^–10^−^^18^ mol/L; (**C**) chloroquine diphosphate 10^−^^19^–10^−^^17^ mol/L; (**D**) lopinavir 10^−^^19^–10^−^^17^ mol/L; (**E**) ribavirin 10^−^^19^–10^−^^17^ mol/L; (**F**) bromhexine 10^−^^21^–10^−^^19^ mol/L; and (**G**) hydroxychloroquine 10^−^^14^–10^−^^12^ mol/L (**H**) KCl 10^−^^15^–10^−^^13^ mol/L as control.

**Figure 7 biosensors-12-00888-f007:**
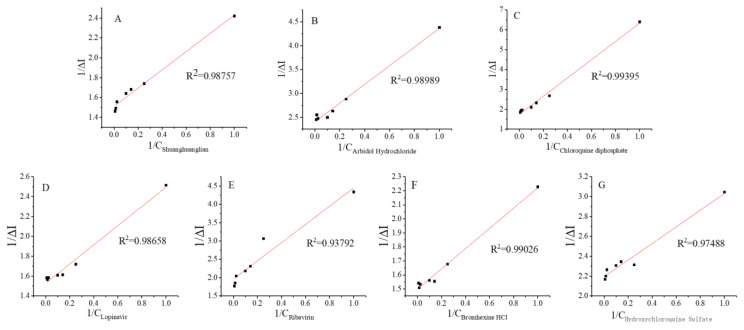
Double reciprocal plots of the fitted response curves of (**A**) Shuanghuanglian oral liquid 10^−^^20^–10^−^^18^ mol/L; (**B**) abridol 10^−^^20^–10^−^^18^ mol/L; (**C**) chloroquine diphosphate 10^−^^19^–10^−^^17^ mol/L; (**D**) lopinavir 10^−^^19^–10^−^^17^ mol/L; (**E**) ribavirin 10^−^^19^–10^−^^17^ mol/L; (**F**) bromhexine 10^−^^21^–10^−^^19^ mol/L; and (**G**) hydroxychloroquine 10^−^^14^–10^−^^12^ mol/L.

**Figure 8 biosensors-12-00888-f008:**
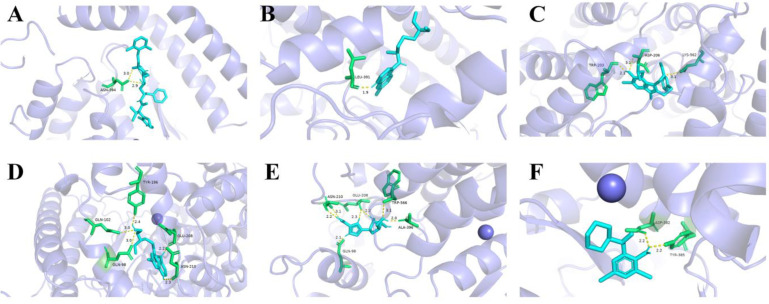
3D schematic diagram of the molecular docking simulation of the spike protein with (**A**) lopinavir, (**B**) chloroquine diphosphate, (**C**) abridol, (**D**) hydroxychloroquine, (**E**) ribavirin, and (**F**) bromhexine.

**Table 1 biosensors-12-00888-t001:** The affinity values of the optimal spike protein/lopinavir binding conformation.

Mode	Affinity (kcal/mol)	Dist From rmsd l.b.	Best Mode rmsd u.b.
1	−8.9	0.000	0.000
2	−8.7	3.837	7.715
3	−8.5	3.917	7.819
4	−8.2	3.689	8.673
5	−8.1	3.782	7.807
6	−8.1	21.586	29.164
7	−8.0	14.650	19.782
8	−7.8	3.704	9.997

**Table 2 biosensors-12-00888-t002:** Molecular docking simulation results and Ka values.

Drug Molecule	Chemical Structure	Ka value	Affinity	Residue Interactions
Bromohexine	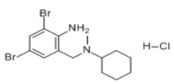	4.823 × 10^−^^22^ mol/L	−5.8 kcal/mol	HB: Tyr385, Asp382 HI: His401, His378, Ala348, Asp350, Thr347
Shuanghuanglian oral liquid	—	6.083 × 10^−^^21^ mol/L	—	—
Arbidol	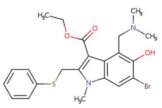	8.140 × 10^−^^21^ mol/L	−5.9 kcal/mol	HB: Asp206, Trp203, Lys562 HI: Tyr202, Asn397, Asn394, Gly205, Glu398, Tyr196
Lopinavir	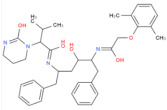	6.325 × 10^−^^20^ mol/L	−8.9 kcal/mol	HB: Asn394 HI: Trp69, Ala99, Leu73, Leu391, Arg393, Phe390, Asp350, Phe40, Trp349, Ala348, His378, His401, Glu402
Ribavirin	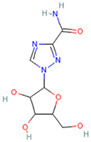	1.270 × 10^−^^19^ mol/L	−7.0 kcal/mol	HB: Ala396, Glu208, Trp566, Gln98, Asn210 HI: Pro565, Val209, Lys562, Leu95
Chloroquine diphosphate	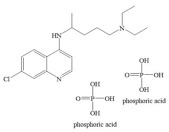	2.644 × 10^−^^19^ mol/L	−4.7 kcal/mol	HB: Leu391 HI: Phe390, Leu73, Lys74, Gln102
Hydroxychloroquine	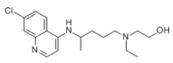	3.825 × 10^−^^15^ mol/L	−6.5 kcal/mol	HB: Tyr196, Glu208, Gln102, Gln98, Asn210 HI: Val209, Lys562, Gly205, Leu95

**Table 3 biosensors-12-00888-t003:** Electrochemical sensor based on SARS-CoV-2 detection with spike protein.

Sensor Type	Analyte	Electrode Type	Minimum Detection Limit	Measurement Method
electrochemical biosensor [29]	spike protein	screen-printed carbon electrode(SPCE)	0.3 fg/mL	differential pulse voltammetry(DPV)
surface-enhanced Raman spectroscopy(SERS) sensor [30]	spike protein RBD	silicon nanowires	9.3 × 10^−^^12^ mol/L	SERS spectroscopy
aptasensor [31]	spike protein RBD	screen-printed carbon electrode	66 pg/mL	electrochemical impedance spectroscopy
electrochemical biosensor [32]	spike protein	glassy carbon electrode and SPCE	1 ag/mL~10 fg/mL	voltammetry
electrochemical immunosensor [33]	spike protein	carbon electrodes (DEP)	0.4 pg/mL for HCoV, 1.0 pg/mL for MERS-CoV	square wave voltammetry (SWV)
SERS-based biosensor [34]	SARS-CoV-2 virus in untreated saliva	silicon wafer	6.07 pg/mL	SERS spectroscopy
electrochemical biosensor combined with recombinase polymerase amplification (RPA) [35]	SARS-CoV-2	multi-microelectrode array	0.972 fg/μL for RdRP gene, 3.925 fg/μL for N gene	DPV
a nucleic acid amplification-free electrochemical biosensor [1]	SARS-CoV-2 RNA	SPCE	5.0 ag/μL for S gene, 6.8 ag/μL for Orf gene	square-wave voltammetry (SWV)
molecularly imprinted polymer-based electrochemical sensor [36]	SARS-CoV-2 nucleoprotein (ncovNP)	thin film electrode	1.5 × 10^−^^14^ mol/L	DPV
electrochemical receptor sensors (this study)	spike protein	glassy carbon electrode	3.3 × 10^−^^20^ mol/L	cyclic voltammetry

## Data Availability

This article has not been submitted to other journals, and the cited materials are labeled references.

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
