# Peer review of "Kinetics of Drug Molecule Interactions with a Newly Developed Nano-Gold-Modified Spike Protein Electrochemical Receptor Sensor"

_biosensors, 2022, doi:10.3390/bios12100888_

Round 1

Reviewer 1 Report

Ref: biosensors-1948378

Title of the manuscript:Kinetics of drug molecule interactions with a newly developed nano-gold-modified spike protein electrochemical receptor sensor

Lu et al. in this article discuss the development of an electrochemical receptor biosensor for SARS-CoV-2 spike protein. The study has been conducted using seven anti-viral drug compounds.  The role of Ka value has an indirect relation with the sensitivity of the sensor wherein bromhexine having the lowest Ka value shows the most sensitivity. The results have been supported using docking studies.

Overall, I find this paper worth publishing and should be considered after addressing the following updates in the manuscript:

1.      English editing and proper sentence formation should be focussed (Line 13-14; Line 120 (I1 and I2) numerals should be subscript; Line 162 (prepared sensor was stored at 4 degrees C above phosphate buffer. What does above mean???); Line 256-257 and many more).

2.      Materials and method section should be rechecked and updated with all necessary materials. (No details about H2O2; NaOH; Thi-Chin and other chemicals show be incorporated).

3.      Figure captions need to be detailed with proper experimental parameters, electrolytes, electrodes, and conditions involved (Figures 1, 2, 3).

4.      In Line 87, “1.0 V-1.0 V window has been specified for activation”. Please check.

5.      What is “Thi-Chi”. There is no proper explanation or introduction of the same and used directly in Section 2.2.2, “it is dropcasted on GCE”. What is the redox control of the Thionine and Chitosan on the GCE for electrochemical response in the potential window? A figure in this respect should be added.

6.      In equation 1, ‘delta I/%’ has been used. What is the meaning of I/% here?

7.      A control of Thionine, Chitosan, nano Au, HRP alone without nano-Au, should be incorporated in Figure 1 or supplementary files for a clear understanding of their electron transfer properties.

8.      Line 145-154, doesn’t account for the electrolyte solution used. Thionine shows a Redox characteristic itself. Authors need to add a figure as a control.

9.      Line 170-177, involves the concentration of drugs redefined: What do 21, 20, and numerals define? Please state clearly.

10.  Figure 2 shows the CV of a. GCE; b. Thi-Chi and so on …. Is it what the authors are trying to mention? Or is it a. GCE; b. GCE/Thi-Chi and so on. No proper clarity for notations is written (else mention it in the Figure captions).

Reviewer 2 Report

Dingqiang Lu et. al., have reported developing a new electrochemical receptor biosensor by using AuNPs-HRP as an electrochemical signal amplification system. As the authors have mentioned, the electrochemical receptor biosensor quantifies the interaction and kinetics between the spike protein and drug molecules and helps with the assessment of receptor-ligand interactions and drug efficacy. This is a well-established report with novelty and significant scientific content and an acceptable 15% similarity match with previously published literature. Following the below comments, authors may enrich the manuscript overall toward the interests of the Biosensors readers.

1.     LOD should be pointed out in the abstract.

2.     The mechanism of action and AuNPs-HRP background should be explained in the introduction.

3.     Present the TEM and SEM micrographs.

4.   In figure3, 4, and 5, add the control.

5.     Expand section 4. Discussion and discuss more results while comparing the finding of this study with the literature.

6.     Section 5. Conclusion is too brief and short. What are the possible application and also suggestions for future work?

Round 2

Reviewer 1 Report

Ref: biosensors-1948378

Title of the manuscript:Kinetics of drug molecule interactions with a newly developed nano-gold-modified spike protein electrochemical receptor sensor

Lu et al. in this article discussed the development of an electrochemical receptor biosensor for SARS-CoV-2 spike protein. The study has been conducted using seven anti-viral drug compounds.  The role of Ka value has an indirect relation with the sensitivity of sensor wherein bromhexine having the lowest Ka value shows the most sensitivity. The results have been supported using docking studies. The writing style needs extensive refining. It should be revised

1.      Even after the English grammatical editing and response to the reviewer, the language used is quite odd. The use of ‘semantic words’ for example- Line 112. “ …. and put a drop on the center of the electrode surface’ can be re-written as “Dropcasted 2 mL of the solution on the working electrode.”

2.      Many spelling mistakes and errors in writing style.

Line 200uL – it's not ‘ uL' rather it should be ‘mL'

There should be proper spacing between the numerals and units.

Similar errors in lines 106-112, 116-119, etc.

Line 122-124, there are so many grammatical errors. “ …. the S protein solution dropwise………….” Should be rewritten as “the S protein solution was added dropwise.”

Similar to the case of Line 227, etc.

3.      Figure 3B, what are 1 and 2 (not mentioned in Figure or its caption)?

4. Equation 1 is ‘deltaI/%’ and why not ‘deltaI%’. Why is extra ‘/’ slash present? Is there any mathematical reasoning?

5.      As asked in the earlier review, the control for Thionin alone, chitosan alone, nano Au alone, and HRP alone have to be supported to ensure no false response in the analysis.

6.      In Figure 3B, the authors mentioned that chitosan/GCE shows response 3B (curve 2). What redox peaks contribute to this response at 0.1 V.

Reviewer 2 Report

1. The author was asked to present the TEM and SEM micrographs and not the AFM.

2. add the control to figures 4, and 5.

3. Add a table to compare the result of this study with previous literature.
